# Haplotypes spanning centromeric regions reveal persistence of large blocks of archaic DNA

**Sasha A Langley[1,2], Karen H Miga[3], Gary H Karpen[1,2], Charles H Langley[4]***

[1]Department of Molecular and Cell Biology, University of California, Berkeley, Berkeley, United States; [2]Biological Systems and Engineering Division, Lawrence Berkeley National Laboratory, Berkeley, United States; [3]UC Santa Cruz Genomics Institute, University of California, Santa Cruz, Santa Cruz, United States; [4]Department of Evolution and Ecology, University of California, Davis, Davis, United States

**Abstract** Despite critical roles in chromosome segregation and disease, the repetitive structure and vast size of centromeres and their surrounding heterochromatic regions impede studies of genomic variation. Here we report the identification of large-scale haplotypes (*cenhaps*) in humans that span the centromere-proximal regions of all metacentric chromosomes, including the arrays of highly repeated α-satellites on which centromeres form. *Cenhaps* reveal deep diversity, including entire introgressed Neanderthal centromeres and equally ancient lineages among Africans. These centromere-spanning haplotypes contain variants, including large differences in α-satellite DNA content, which may influence the fidelity and bias of chromosome transmission. The discovery of *cenhaps* creates new opportunities to investigate their contribution to phenotypic variation, especially in meiosis and mitosis, as well as to more incisively model the unexpectedly rich evolution of these challenging genomic regions.
DOI: https://doi.org/10.7554/eLife.42989.001

*For correspondence:
chlangley@ucdavis.edu

**Competing interests:** The authors declare that no competing interests exist.

## Introduction

The centromere is the unique chromosomal locus that forms the kinetochore, which interacts with spindle microtubules and directs segregation of replicated chromosomes to daughter cells (*McKinley and Cheeseman, 2016*). Human centromeres assemble on a subset of large blocks (many Mbps) of highly repeated (171 bp) α-satellite arrays found on all chromosomes. These repetitive arrays and the flanking heterochromatin (together the Centromere Proximal Regions or CPRs) play critical roles in the integrity of mitotic and meiotic inheritance (*Janssen et al., 2018*). In somatic tissues, chromosome instability, including loss and gain of chromosomes, plays large and complex roles in aging, cancer (*Naylor and van Deursen, 2016*), and human embryonic survival (*McCoy, 2017*). Sequence variation in CPRs can affect meiotic pairing (*Dernburg et al., 1996*; *Karpen et al., 1996*), kinetochore formation (*Rosin and Mellone, 2017*) and nonrandom segregation (*Karpen et al., 1996*). A large component of genetic disease stems from aneuploidies arising during meiosis (*Nagaoka et al., 2012*). Further, the unique asymmetry of transmission in female meiosis, where only one parental chromosome is transmitted, presents an opportunity for the evolution of strong deviations from mendelian segregation ratios (meiotic drive) (*Pardo-Manuel de Villena and Sapienza, 2001*; *Chmátal et al., 2014*). Recurrent meiotic drive is a potential cause of the evolutionarily rapid divergence of satellite DNAs and centromeric chromatin proteins (*Malik and Henikoff, 2001*; *Talbert et al., 2004*; *Rudd et al., 2006*; *Malik and Henikoff, 2009*), as well as the observed high levels of meiotic aneuploidy arising from the tradeoff between fidelity and drive

(*Zwick et al., 1999*). However, the challenges inherent to assessing genomic variation in these repetitive and dynamic regions remain a significant barrier to incisive functional and evolutionary investigations.

## Results

Recognizing the potential research value of well-genotyped diversity across human CPRs, we hypothesized that the low rates of meiotic exchange in these regions (*Nambiar and Smith, 2016*) might result in large haplotypes in populations, perhaps even spanning the α-satellite arrays. To test this, we examined the Single Nucleotide Polymorphism (SNP) linkage disequilibrium (LD) and haplotype variation surrounding the centromeres among the diverse collection of genotyped individuals in Phase 3 of the 1000 Genomes Project (*Auton et al., 2015*). *Figure 1a* depicts the predicted patterns of strong LD (red) and associated unbroken haplotypic structures surrounding the gap of unassembled satellite DNA of a metacentric chromosome. Unweighted Pair Group Method with Arithmetic Mean (UMPGA) clustering on 800 SNPs immediately flanking the chrX centromeric gap in males (*Figure 1c*) reveals a clear haplotypic structure that spans the gap and extends, as predicted, to a much larger region (≈7 Mbp, *Figure 1b*). Similar clustering of the imputed genotypes of females also falls into the same distinct high-level haplotypes (*Figure 1—figure supplement 1*). This discovery of the predicted haplotypes spanning CPRs (hereafter referred to as *cenhaps*) on chrX and most metacentric chromosomes (*Figure 2*) opens a new window into their evolutionary history and functional associations.

The pattern of geographic differentiation across the inferred chrX CPR (*Figure 1b,c*) exhibits higher diversity in African samples, as observed throughout the genome (*Auton et al., 2015*). Despite being fairly common among Africans today, a distinctly diverged chrX cenhap (cenhap 1, highlighted in purple, *Figure 1b,c*) is rare outside of Africa. Examination of the haplotypic clustering and estimated synonymous divergence in the coding regions of 21 genes included in the chrX cenhap region (see *Figure 1—source data 1*) yields a parallel relationship among the three major cenhaps and an estimated Time of the Most Recent Common Ancestor (TMRCA) of ≈600 KYA (*Figure 1d*) for this most diverged example. While ancient segments have been inferred in African genomes (*Hammer et al., 2011*; *Hsieh et al., 2016*), this cenhap stands out as genomically (if not genetically) large. The presence of such polymorphic ancient cenhaps is inconsistent with the predicted hitchhiking effect of sequential fixation of new meiotically driven centromeres (*Malik and Henikoff, 2001*). Further, the detection of near-ancient segments spanning the centromere contrasts with the observation of substantially more recent ancestry across the remainder of chrX and with the expectation of reduced archaic sequences on chrX (*Dutheil et al., 2015*). The large block on the right in *Figure 1b* is comprised of SNPs in exceptionally high frequency in Africans. The synonymous divergence in coding genes in this block indicates it too is quite old (data not shown) and may share ancestry with the ancient African cenhap. Putative distal recombinants of this block are observed outside of Africa and may contribute to associations of SNPs in this region with a diverse set of phenotypes, including male pattern hair loss (*Hagenaars et al., 2017*) and prostate cancer (*Al Olama et al., 2014*).

This deep history of the chrX CPR raises the possibility of even more ancient lineages on other chromosomes, either derived by admixture with archaic hominins or maintained by balancing forces. Although putatively introgressed archaic segments in African genomes have been inferred from genome-wide demographic modeling (*Hammer et al., 2011*; *Hsieh et al., 2016*; *Durvasula and Sankararaman, 2019*; *Speidel et al., 2019*), ancient cenhaps could also persist within the ancestral population due to natural selection. The relatively recent origin of AMHs outside of Africa and the availability of Neanderthal and Denisovan genomes derived from fossil DNAs support more direct methods for detection of enrichment of archaic segments outside of Africa (*Green et al., 2010*). Such studies firmly establish genome-wide evidence of recent introgression into Eurasian populations of AMHs (*Green et al., 2010*; *Patterson et al., 2012*; *Sankararaman et al., 2012*; *Prüfer et al., 2014*; *Sankararaman et al., 2014*; *Prüfer et al., 2017*).

To identify likely Neanderthal or Denisovan introgressed cenhaps, we looked for highly diverged examples in non-African populations (see *Figure 2*) that shared a strong excess of derived alleles with those archaic hominids and not with African genomes, using chimpanzee as the outgroup (*Green et al., 2010*; *Patterson et al., 2012*; *Prüfer et al., 2014*; *Prüfer et al., 2017*). Applying this

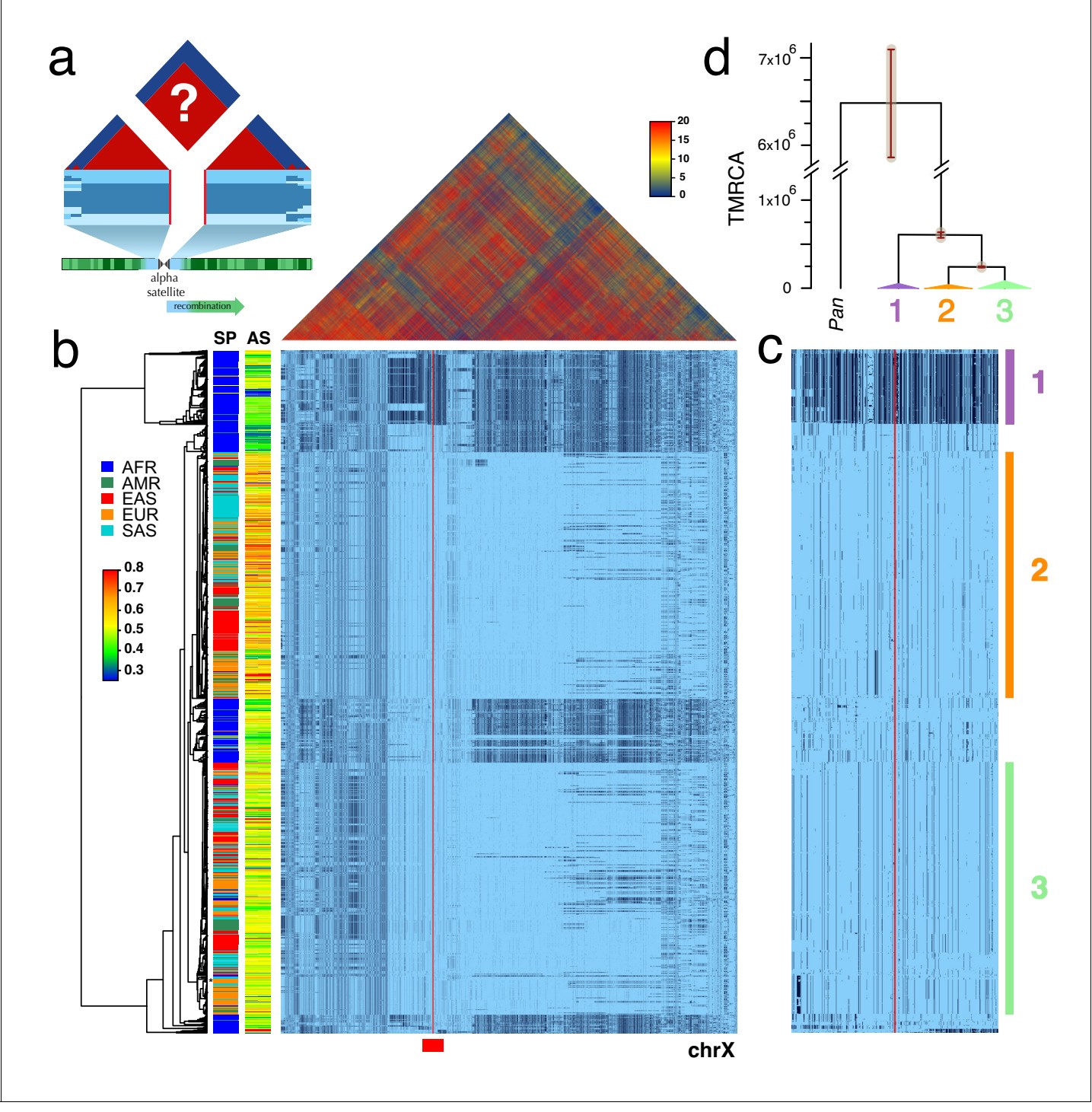

**Figure 1.** Strong LD across centromeric gaps forms large-scale centromere-spanning haplotypes, or *cenhaps*. A full resolution version of this figure is available as *Figure 1—source data 2*. (**a**) The predicted patterns of the magnitude of linkage disequilibrium (LD) (triangle at top) for a Centromere Proximal Region (CPR) in a metacentric human chromosome (bottom) in a large outbreeding population. Central blue bands represent clustered haplotypes expected if crossing-over declines to zero in and around the highly repeated α-satellite DNA (central assembly gap) and the SNP-rich flanking regions (light blue). (**b**) Triangle (top) shows the LD between pairs of 17702 SNPs (Left: chrX:55623011–58563685, Right: chrX:61725513–68381787; hg19) flanking the centromere and α-satellite assembly gap (red vertical line) from 1231 human male X chromosomes from the 1000 Genomes Project. The color maps (see adjacent legend) to the -$log_{10}(p)$ where the $p$ value derives from the 2×2 $\chi^2$ for independence of alleles at each pair of SNPs. Below, a broad haplotypic representation of these same data. SNPs were filtered for minor allele count (MAC) $\geq$ 60, but not by *4gt_dco*. Minor alleles shown in black. Poorly genotyped SNPs near edges of the gap (red line) were masked. Superpopulation (**SP**; **AFR**ica, **AM**e**R**icas, **E**ast **AS**ia,

*Figure 1 continued on next page*

*Figure 1 continued*

EURope, South ASia) and scaled estimate of chrX-specific α-satellite array size (AS) indicated at left side. Approximate position of HuRef chrX indicated by black asterisk at right of the tree. Dendrogram represents UPGMA clustering based on the hamming distance between haplotypes comprised of 800 filtered SNPs immediately flanking the centromere (Left: chrX:58374895–58563685, Right: chrX:61725513–61921419; hg19), indicated by red bar at bottom and shown in detail in **c**. The three most common X cenhaps are highlighted with colored vertical bars. (**d**) A UPGMA tree based on the synonymous divergence in 21 genes (see *Figure 1—source data 1*) in the three major chrX cenhaps (indicated in **c**), assuming the TMRCA of humans and chimps is 6.5MY. The bars at each node represent ±two standard deviations of distributions of estimated TMRCAs across the genes. Widths of the triangles are proportional to the $log_{10}$ of number of members of each cenhap, and the height is proportional to the average divergence within each cenhap.

DOI: https://doi.org/10.7554/eLife.42989.002

The following source data and figure supplements are available for figure 1:

**Source data 1.** The 21 chrX coding genes in the CPR (8 left and 13 right of the centromere gap) used in the UPGMA clustering and estimation of TMRCAs.
DOI: https://doi.org/10.7554/eLife.42989.007
**Source data 2.** Full resolution version of *Figure 1*.
DOI: https://doi.org/10.7554/eLife.42989.008
**Figure supplement 1.** X chromosome cenhaps from phased female data align with those from haploid males.
DOI: https://doi.org/10.7554/eLife.42989.003
**Figure supplement 1—source data 1.** Full resolution version of *Figure 1—figure supplement 1*.
DOI: https://doi.org/10.7554/eLife.42989.004
**Figure supplement 2.** Filtering of chrX CPR recombinants for CDS divergence, expected heterozygosity and TMRCAs.
DOI: https://doi.org/10.7554/eLife.42989.005
**Figure supplement 2—source data 1.** Full resolution version of *Figure 1—figure supplement 2*.
DOI: https://doi.org/10.7554/eLife.42989.006

approach to the CPR of chr11 revealed a compelling example of Neanderthal introgression, which is illustrated in *Figure 3a* in the context of the seven most common chr11 cenhaps. The most diverged lineage contains a small basal group of primarily out-of-Africa genomes (cenhap 1, highlighted in green). This cenhap carries a large proportion of the derived alleles assigned to the Neanderthal lineage, DM/(DM+DN)=0.98, where DM is the cenhap mean number of shared Neanderthal Derived Matches and DN is the cenhap mean number of Neanderthal Derived Non-matches (*Figure 3a*, at left). The ratio DM/(DM+AN)=0.91, where AN is the number of Neanderthal-cenhap Non-matches that are Ancestral in the Neanderthal, suggests that this large cenhap lineage, including the centromere and satellite sequences, shared most of its evolutionary history with Neanderthals. This diverged cenhap is limited to populations outside of Africa, supporting the conclusion that it is an introgressed archaic centromere. *Figure 3b* shows these mean counts for each SNP class by cenhap group, confirming that the affinity to Neanderthals is slightly stronger than to Denisovans. A second basal lineage found principally in Africa (cenhap 2, highlighted in purple, *Figure 3a*) separates shortly after the inferred Neanderthal. It is unclear if this cenhap represents an introgression from a distinct archaic hominin in Africa or a surviving ancient lineage within the population that gave rise to AMHs.

The relatively large expanses of these chr11 cenhaps and unexpectedly sparse evidence of recombination could be explained by either relatively recent introgressions or cenhap-specific suppression of crossing over with other AMH genomes in this CPR (e.g., an inversion). As with chrX above, the clustering of cenhaps based on coding synonymous SNPs in an extended window containing 37 genes (*Figure 3d*, *Figure 3—figure supplement 1*) yields a congruent topology and estimates of TMRCAs for the two basal cenhaps of ~1 and ~0.5 MYA, consistent with relatively ancient origins. Among the 37 genes in the chr11 CPR are 34 of the ~300 known (*Malnic et al., 2004*) odorant receptors (ORs). Nonsynonymous SNPs in these chr11 ORs are associated with variation in human olfactory perception of particular volatile chemicals (*Trimmer et al., 2019*). 73 amino acid replacements polymorphisms are observed in 25 of these ORs (*Figure 3—source data 1*). The vast majority (63) are on the lineages to the two putative archaic cenhaps. Indeed, 60 are on the lineage to the Neanderthal haplotype suggesting this cenhap encodes Neanderthal-specific determinants of smell and taste. Similarly, in the second putative archaic African cenhap, seven of these ORs harbor ten amino acid replacements, of which only one is shared with cenhap 1 (see *Figure 3—source data 1*). The frequencies of the Neanderthal cenhap in Europe, South Asia and the Americas (0.061, 0.032

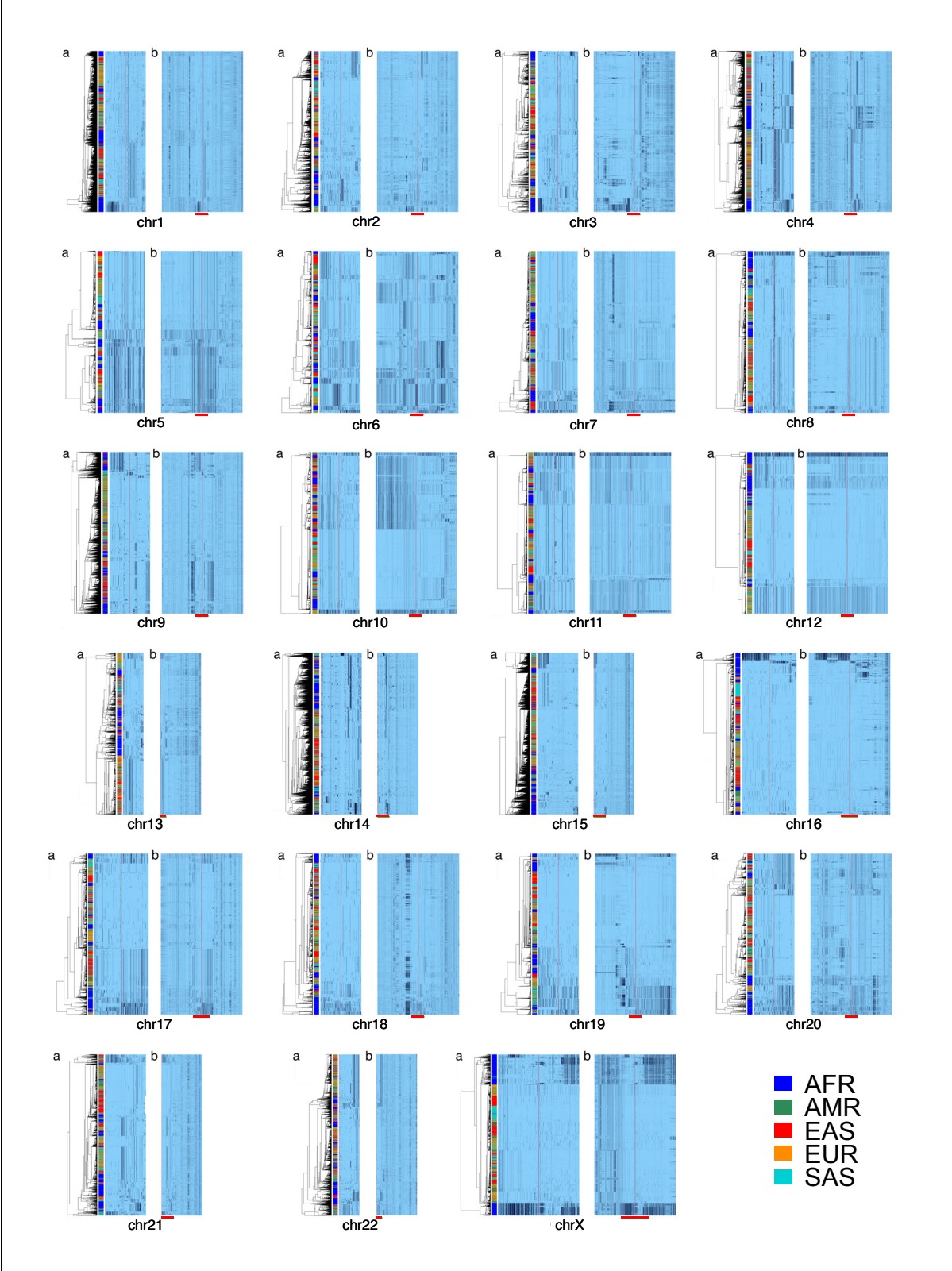

**Figure 2.** Cenhap diversity is found on many chromosomes. A full resolution version of this figure is available as *Figure 2—source data 2*. SNPs were filtered for MAC ≥ 80 and passing the *4gt_dco* with a tolerance of 0 (see Materials and methods). Minor alleles shown in black, assembly gap is indicated by red line. Panel (a) for each chromosome shows the diversity in a subset of SNPs immediately surrounding the gap. SNPs from panel **a** were used for UPGMA clustering based on the hamming distance (see Materials and methods and *Figure 2—source data 1*). Panel (**b**) for each

*Figure 2 continued on next page*

*Figure 2 continued*

chromosome is the haplotypic representation of SNPs in the CPR of each chromosome based on imputed genotypes from the 1000 Genomes Project (see *Figure 2—source data 1* for coordinates), using the clustering as for panel (a). The red bar at the bottom of panel (b) shows the position of the clustering region depicted in (a). Superpopulation is indicated in bar at far left.

DOI: https://doi.org/10.7554/eLife.42989.009

The following source data is available for figure 2:

**Source data 1.** Centromere-Proximal Regions examined.
DOI: https://doi.org/10.7554/eLife.42989.010
**Source data 2.** Full resolution version of *Figure 2*.
DOI: https://doi.org/10.7554/eLife.42989.011

and 0.033 respectively), and of second ancient cenhap in Africa (0.036), are sufficiently high that together they contribute $\approx$ 18% of the amino acid replacement diversity in these 34 ORs among the 1000 Genomes. Thus, a substantial and rather unique part of the variation in chemical perception among AMH may be contributed by these two ancient cenhaps.

The most diverged, basal clade in the chr12 CPR (*Figure 3c*, indicated in brown) is common in Africa, but, like the most diverged chrX cenhap, is not represented among the descendants of the out-of-Africa migrations (*Bae et al., 2017*). The great depth of the lineage of this cenhap is further supported by comparison to homologous archaic sequences (*Green et al., 2010*; *Prüfer et al., 2014*; *Prüfer et al., 2017*). Consistent with the hypothesis that this branch split off before that of Neanderthals/Denisovans, members of this cenhap share fewer matches with derived SNPs on the Neanderthal and Denisovan lineages (DM) and exhibit strikingly more ancestral non-matches (AN) than other chr12 cenhaps (see *Figure 3b*). This putatively archaic chr12 cenhap represents a large and obvious example of the potentially introgressed sequences within African populations inferred from model-based analyses of the distributions of sequence divergence (*Hammer et al., 2011*; *Hsieh et al., 2016*; *Durvasula and Sankararaman, 2019*). The small out-of-Africa cenhap nested within a mostly African subclade (indicated in blue in *Figure 3c*) appears to be a typical Eurasian archaic introgression with higher affinity to Neanderthals (DM/(DN + DM)=0.91 and DM/(DM +AN) =0.90) than to Denisovans (*Figure 3b*). This bolsters the conclusion that the basal African cenhap represents a distinctly older archaic lineage. Unfortunately, there are too few coding bases in this region to support confident estimation of the TMRCAs of these ancient chr12 cenhaps. Based on the numbers of SNPs underlying the cenhaps, this basal cenhap is twice as diverged as the apparent introgressed Neanderthal cenhap, placing the TMRCA at ~1.1 MYA, assuming the Neanderthal TMRCA was 575KYA (*Prüfer et al., 2017*). While there is no direct evidence of recent introgression, the large genomic scale of the most diverged chr12 cenhap (relative to apparent exchanges in other cenhaps) is consistent with recent admixture with an extinct archaic in Africa; yet, again, selective maintenance of ancient cenhaps with associated suppression of crossing over is an alternative explanation.

Chromosomes X, 11 and 12 harbor a diversity of large cenhaps, including those representing archaic lineages. Notably, the CPRs of other chromosomes include diverged, basal lineages that are likely to be relatively old, if not archaic (*Figure 2*). Two examples are chromosome 8, containing an ancient cenhap limited to Africa with an estimated TMRCA of ~730 KYA (*Figure 3—figure supplement 2*), and chr10 that appears to harbor another clear Neanderthal cenhap introgression (*Figure 3—figure supplement 3*). Obtaining genomic sequence from African archaics would shed light on the evolutionary origins of the ancient cenhaps not associated with Neanderthal and Denisovan introgression. It should be noted that the very large genomic sizes of these ancient cenhaps could allow identification of archaic homology even with modest genomic sequence coverages from archaic fossils.

These SNP-based cenhaps portray a rich, highly structured view of the diversity in the unique segments flanking repetitive regions. While the divergence of satellites may be dynamic on a shorter time scale (*Smith, 1976*), we reasoned that the paucity of exchange in these regions would create cenhap associations with satellite divergence in both sequence and array size. *Miga et al. (2014)* generated chromosome-specific graphical models of the $\alpha$-satellite arrays and reported a bimodal distribution in estimated chrX-specific $\alpha$-satellite array (DXZ1) sizes (*Willard et al., 1983*) for a subset

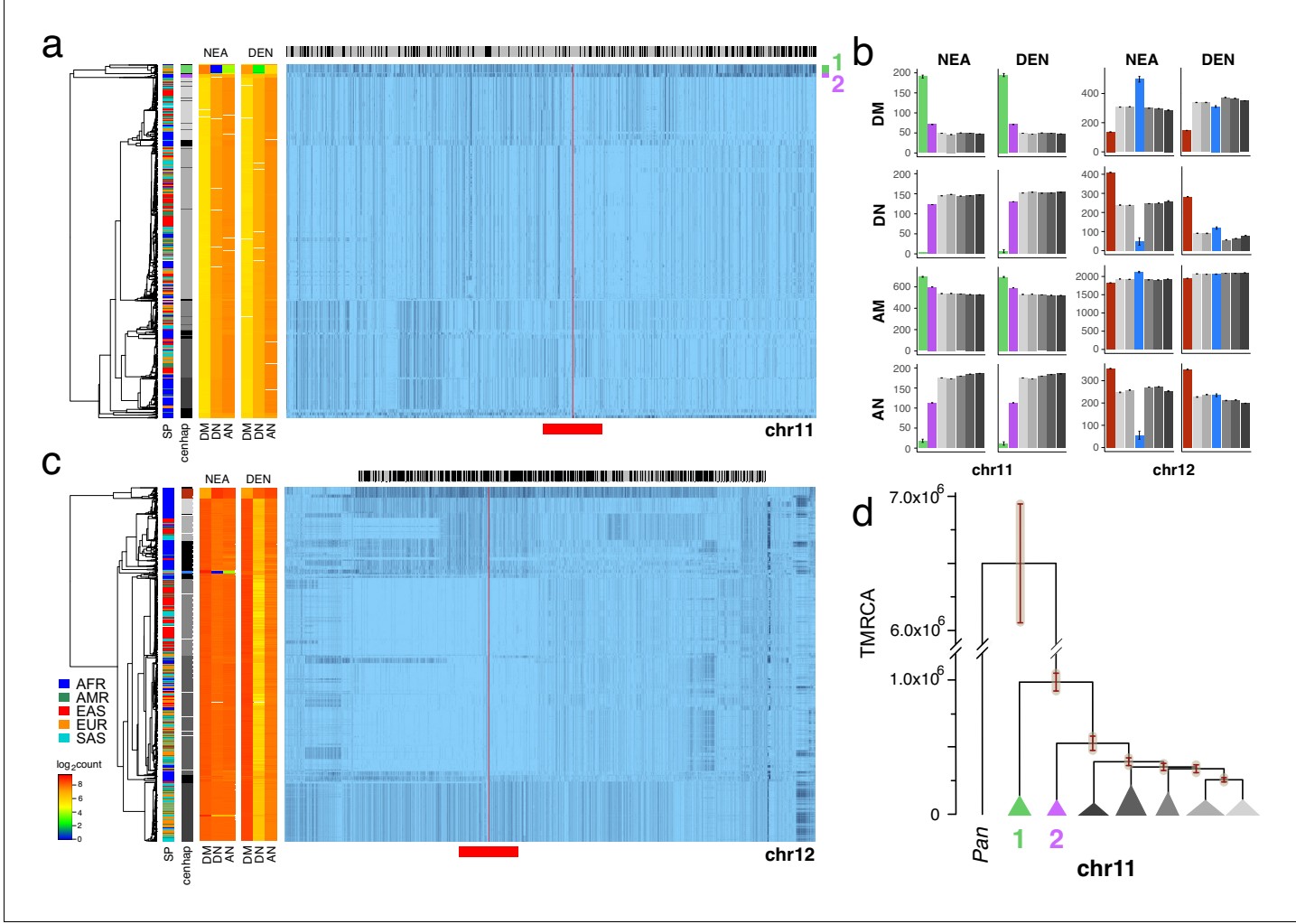

**Figure 3.** Archaic cenhaps are found in AMH populations. A full resolution version of this figure is available as *Figure 3—source data 3*. (a) Haplotypic representation of 8816 SNPs from 5008 imputed chr11 genotypes from the 1000 Genomes Project (Left: chr11:50509493–51594084, Right: chr11:54697078–55326684; hg19). SNPs were filtered for MAC ≥ 35 and passing the *4gt_dco* with a tolerance of three (see Materials and methods). Minor alleles shown in black and assembly gap indicated by red line. Haplotypes were clustered with UPGMA based on the hamming distance between haplotypes comprised of 1000 SNPs surrounding the gap (Left: chr11:51532172–51594084, Right: chr11:54697078–54845667; hg19, indicated by red bar at bottom). Superpopulation and cenhap partitioning are indicated by bars at far left. Log$_2$ counts of DM (derived in archaic, shared by haplotype), DN (derived in archaic, not shared by haplotype) and AN (ancestral in archaic, not shared by haplotype) for each cenhap relative to Altai Neanderthal (NEA) and Denisovan (DEN) at left. Gray horizontal bar (top) indicates region included in analysis of archaic content; black bars indicate SNPs with data for archaic and ancestral states. (b) Bar plots indicating the mean and 95% confidence intervals of DM, DN, AM (ancestral in archaic, shared by cenhap) and AN counts for cenhap groups (as partitioned in a. and c.) relative to Altai Neanderthal and Denisovan genomes, using chimpanzee as an outgroup (*Speidel et al., 2019*). (c) Haplotypic representation, as above, of 21950 SNPs from 5008 imputed chr12 genotypes from the 1000 Genomes Project (Left: chr12:33939700–34856380, Right: chr12:37856765–39471374; hg19). SNPs were filtered for MAC ≥ 35. Haplotypes were clustered with UPGMA based on 1000 SNPs surrounding the gap (Left: chr12:34821738–34856670, Right: chr12:37856765–37923684; hg19). Bars at side, top and bottom same as in a. (d) A UPGMA tree based on the synonymous divergence for 30 genes in the seven major chr11 cenhaps (see *Figure 3—source data 2*), assuming the TMRCA of humans and chimpanzee is 6.5MY (see Materials and methods and legend for *Figure 1d*). The error bars at each node represent ±two standard deviations of distributions of estimated TMRCAs across the genes.

DOI: https://doi.org/10.7554/eLife.42989.012

The following source data and figure supplements are available for figure 3:

**Source data 1.** The 37 chr11 coding genes in the CPR (2 left and 35 right of the centromere gap) used in the UPGMA clustering and estimation of TMRCAs.

DOI: https://doi.org/10.7554/eLife.42989.019

**Source data 2.** The eight chr8 coding genes in the CPR (8 left and 0 right of the centromere gap) used in the UPGMA clustering and estimation of TMRCAs.

*Figure 3 continued on next page*

*Figure 3 continued*

DOI: https://doi.org/10.7554/eLife.42989.020

**Source data 3.** Full resolution version of *Figure 3*.

DOI: https://doi.org/10.7554/eLife.42989.021

**Figure supplement 1.** Region of chromosome 11 used for cenhap coding region divergence.

DOI: https://doi.org/10.7554/eLife.42989.013

**Figure supplement 1—source data 1.** Full resolution version of *Figure 3—figure supplement 1*.

DOI: https://doi.org/10.7554/eLife.42989.014

**Figure supplement 2.** Evidence of an archaic cenhap within Africa on chromosome 8.

DOI: https://doi.org/10.7554/eLife.42989.015

**Figure supplement 2—source data 1.** Full resolution version of *Figure 3—figure supplement 2*.

DOI: https://doi.org/10.7554/eLife.42989.016

**Figure supplement 3.** Evidence of archaic cenhap introgression on chromosome 10.

DOI: https://doi.org/10.7554/eLife.42989.017

**Figure supplement 3—source data 1.** Full resolution version of *Figure 3—figure supplement 3*

DOI: https://doi.org/10.7554/eLife.42989.018

of the 1000 Genomes males. *Figure 1b* extends this observation to the entire data. The cumulative distributions of estimated array sizes of the three common chrX cenhaps designated in *Figure 1c* show substantial differences (*Figure 4a*). α-satellite array sizes in cenhap-homozygous females are parallel to males, and imputed cenhap heterozygotes are intermediate, as expected. Similarly, *Figure 4b* shows an even more striking example of variation in array size between cenhap homozygotes on chr11, and *Figure 4c* demonstrates that heterozygotes of the two most common cenhaps are reliably intermediate in size. While we confirmed that reference bias does not explain the observed cenhaps with large array size on chrX and chr11 (see Materials and methods, *Figure 1b*, *Figure 4b* and *Figure 4—figure supplement 1*), it is a potential explanation for particular instances of cenhaps with small estimated array sizes, for example the relatively low chrX-specific α-satellite content in the highly diverged African cenhap (see *Figure 1b,c* and *Figure 4a*, cenhap 1, highlighted in purple). Importantly, our results demonstrate that cenhaps do robustly tag a substantial component of the genetic variation in array size.

## Discussion

The potential impact of sequence variation in CPRs and their associated satellites on centromere and heterochromatin functions has been long recognized but difficult to study (*Pardo-Manuel de Villena and Sapienza, 2001*). Both binding of the centromere-specific histone, CENPA (*Sullivan et al., 2011*), and kinetochore size (*Iwata-Otsubo et al., 2017*) are known to scale with the size of arrays and to fluctuate with sequence variation in satellite DNAs (*Aldrup-MacDonald et al., 2016*). Through these interactions with kinetochore function and other roles for heterochromatin in chromosome segregation (*Dernburg et al., 1996*; *Karpen et al., 1996*; *Peng and Karpen, 2008*), α-satellite array variations can affect mitotic stability in human cells (*Sullivan et al., 2017*), as well as meiotic drive systems in the mouse (*Chmátal et al., 2014*). Meiotic drive has been proposed as the likely explanation for the saltatory divergence of satellite sequences and the excess of nonsynonymous divergence of several centromere proteins, some of which interact directly with the DNA (*Malik and Henikoff, 2001*; *Talbert et al., 2004*; *Malik and Henikoff, 2009*). However, the high levels of haplotypic diversity and deep cenhap lineages observed (*Figure 2*) conflict with the predictions of a naïve turnover model based on recurrent strong directional selection yielding sequential fixation of new centromeric haplotypes. Indeed, the levels of synonymous diversity, $\pi_s$, in the few coding genes in the CPR of chrX, 0.00062 (0.00043–0.00128) and chr11, 0.00128 (0.00088–0.00217), are not different from levels of diversity in non-CPR regions (*Dutheil et al., 2015*). The genes in the CPR of chr8 show a considerably lower mean $\pi_s$, 0.00010 (0.00007–0.00019); but we note there are only eight genes and their mean divergence from *Pan* orthologs is also low (*Figure 3—source data 2*). The inherent frequency-dependence of meiotic drive (*Charlesworth and Hartl, 1978*), associative overdominance (*Ohta, 1971*), a likely tradeoff between meiotic transmission bias and the fidelity of segregation of driven centromeres (*Zwick et al., 1999*), and the expected impact of unlinked

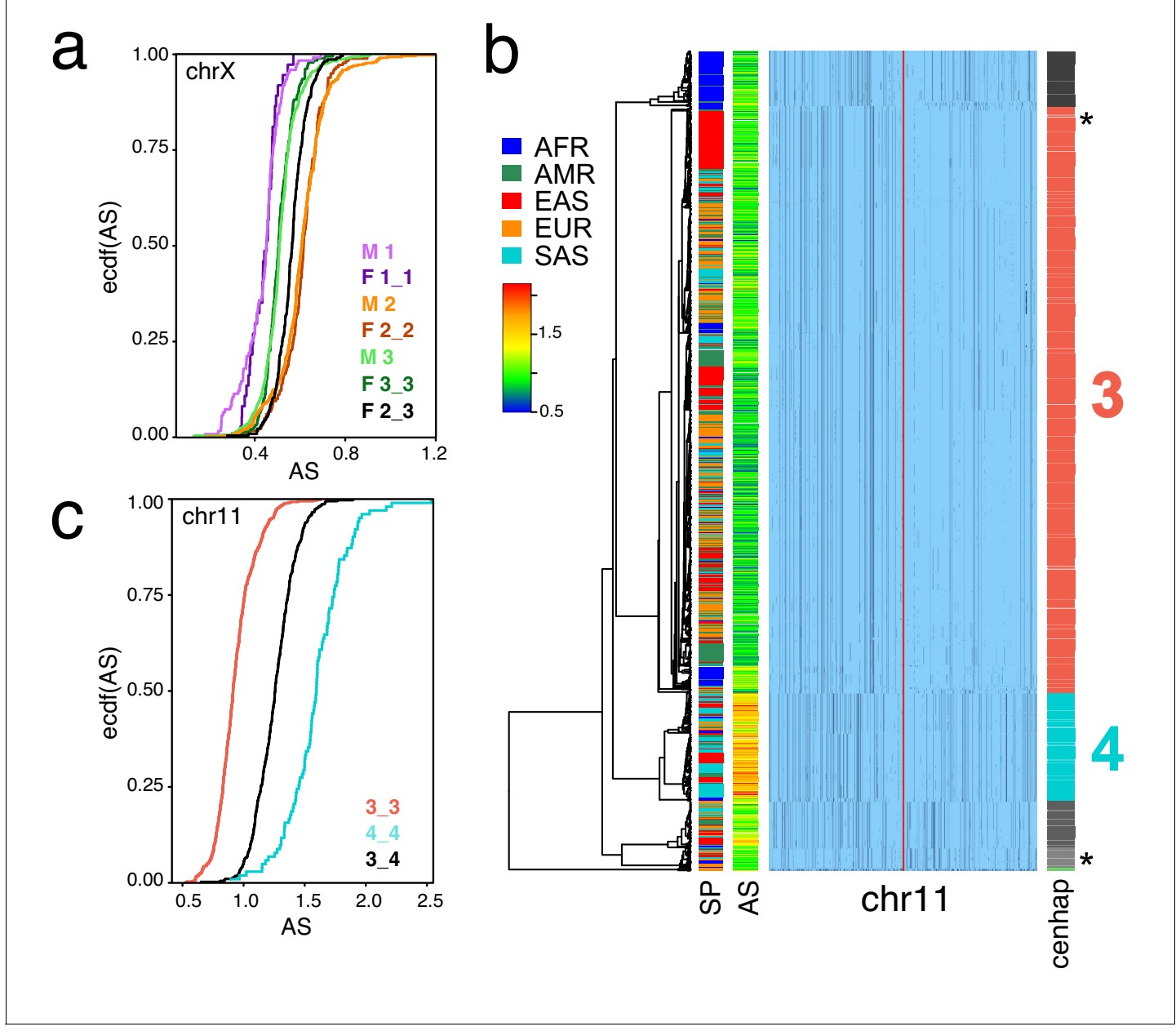

**Figure 4.** Cenhaps differ in α-satellite array size. A full resolution version of this figure is available as *Figure 4—source data 1*. (a) Empirical cumulative densities (ecdf) of chrX α-satellite array size for cenhap homozygotes and heterozygotes (see *Figure 1b* for cenhap designations). 1_2 and 1_3 heterozygotes were excluded due to insufficient data. Female (F) values were normalized (x 0.5) to facilitate plotting with hemizygote male (M) data. (b) Haplotypic representation of 1000 SNPs from 1546 imputed chr11 genotypes from 773 cenhap-homozygous individuals. SNPs were filtered for MAC ≥ 35 and passing the *4gt_dco* with a tolerance of 3. Minor alleles shown in black. Assembly gap indicated by red line. Superpopulation (SP) and scaled chr11-specific α-satellite array size (AS) at left. Cenhap partitions at right; most common cenhap '3' and cenhap with larger mean array size '4' are highlighted. Most probable HuRef cenhap genotypes are indicated by black asterisks at right. (c) Empirical cumulative density of array size for chr11 cenhap (from b) homozygotes (3_3 and 4_4) and heterozygotes (3_4).

DOI: https://doi.org/10.7554/eLife.42989.022

The following source data and figure supplements are available for figure 4:

**Source data 1.** Full resolution version of *Figure 4*.
DOI: https://doi.org/10.7554/eLife.42989.025

**Figure supplement 1.** HuRef's chr11 cenhap genotype.
DOI: https://doi.org/10.7554/eLife.42989.023

**Figure supplement 1—source data 1.** Full resolution version of *Figure 4—figure supplement 1*.

*Figure 4 continued on next page*

*Figure 4 continued*

DOI: https://doi.org/10.7554/eLife.42989.024

suppressors (*Charlesworth and Hartl, 1978*) are plausible forces that would mitigate the impact of hitchhiking and background selection on the levels of standing polymorphism in CPRs.

The identification of human cenhaps raises new questions about the evolution of these unique genomic regions, but also provides the resolution and framework necessary to quantitatively address them. Our results transform large, previously obscure and shunned genomic regions into genetically rich and tractable resources, revealing unexpected diversity, including immense ancient CPRs, several of which are apparent Neanderthal introgressions. Most importantly, cenhaps can now be investigated for associations with variation in evolutionarily important chromosome functions, such as meiotic drive (*Meyer et al., 2012*) and recombination (*Nambiar and Smith, 2016*), as well as disease-related functions, such as aneuploidy in the germline (*Nagaoka et al., 2012*) and in development (*McCoy, 2017*), cancer and aging (*Naylor and van Deursen, 2016*).

## Materials and methods

### Identification of cenhaps in 1000 genomes

SNPs from the 1000 Genomes (Phase 3) (*Auton et al., 2015*) data (ftp.1000genomes.ebi.ac.uk/vol1/ftp/phase3/data/) were examined for linkage disequilibria and haplotypic structure in the Centromere-Proximal Regions of each chromosome (see *Figure 2—source data 1*).

### *4gt_dco* filter of putative genotyping errors

While the imputation used in 1000 Genomes (Phase 3) typically yielded calls in the CPR that fit clearly into the cenhaps, occasionally, in particular regions on particular chromosomes, the called haplotypes appear random (do not associate with the larger scale haplotypes). To filter such SNPs, we applied the *4gt_dco* algorithm base on the following rationale. In a region of very low exchange, genotyping errors can appear as apparent gene conversions at a single site (double cross overs) in the context of a sample comprised of clear haplotypes. In a set of homologous genomic sequences, randomly sampled from an outbreeding diploid population, the equilibrium scale of linkage disequilibrium is approximately $1/4N(r + g)$ (where $N$ is the diploid population size and $r$ is the rate of crossing over, and $g$ is the rate of gene conversion, see *Song et al. (2007)*. If we assume a selectively neutral infinite sites model and that genomic scale of crossing over is much larger than that of gene conversion ($r \ll g$, and the gene conversion track length is also small), both gene conversion AND genotyping errors can be inferred based on a simple test, the observation of all four gametotypes at two linked loci. *4gt_dco* is positive and the focal SNP is filtered, if all four possible two-locus-two-allele gametes between the focal SNP and either of a pair flanking (on opposite sides) SNPs are observed, while the flanking SNPs do not exhibit all four gametes between themselves. We applied *4gt_dco* across the target genomic regions from 5' to 3' in successively larger windows of flanking SNPs of surrounding surviving SNPs. In preliminary analyses (results not shown) we found that applying *4gt_dco* first in a window of ±10, then ±20, ±30 and finally ±40, flanking SNPs (as yet unfiltered) can eliminate SNPs that are not well represented in centromeric haplotypes (cenhaps). We also found that incorporating 'tolerance' (maximum number of pairs of flanking SNPs in a window failing before filtering the focal SNP) improved the performance of *4gt_dco*. On the X chromosome, the data were too sparse to support the *4gt_dco* test. Instead, small regions of contiguous unreliable genotyping at the edges of the assembly flanking the centromeric gap were hard masked (chrX:5856368–61725513). For chromosomes 8, 10 and 11 we applied the *4gt_dco* with a tolerance of 3.

### Haplotype clustering and visualization

To examine haplotypic structure of CPRs in the filtered 1000 Genomes data, we used UPGMA cluster analysis based on the hamming distance of haplotypes comprised of the indicated central subsets of SNPs flanking the assembly gap. Resultant dendrograms were cut to generate cenhap

groups. In some instances, dendrograms were cut at multiple heights to isolate groups of interest. Haplotypic representations were plotted in R using the gplots package.

## Inference of introgression of Neanderthal and/or Denisovan cenhaps

The boundaries of highly diverged cenhaps were determined by excluding flanking regions with apparent exchanges in the history of the 1000 Genomes. 1000 Genomes SNPs (MAC $\geq$ 35 and passing the *4gt_dco* with a tolerance of three) from cenhap regions were classified relative to Altai Neanderthal and Denisovan assemblies (using *Pan troglotytes* as an outgroup) as DM (derived in archaic, match in the imputed haplotype), DN (derived in archaic, no match in haplotype), AM (ancestral in archaic and matching the haplotype) and AN (ancestral in archaic, no match in haplotype, that is derived in the haplotype) (*Prüfer et al., 2017*). Bar plots, generated using ggplot2, depict the mean and standard error of each class for cenhap groups.

## Estimates of cenhap divergence

For estimates of synonymous and nonsynonymous divergence between and diversity, $\pi_s$, within cenhaps (as well as divergence from *Pan*), coding sequences of genes located in the central regions of cenhaps (i.e., where there is little or no evidence of exchange in the descent) were chosen for analyses. We identified the transcript with the longest CDS for each coding cenhap gene in Gencode Release 27 (*Harrow et al., 2012*) annotations and extracted multi-fasta files for these CDS regions from the 1000 Genomes. The corresponding Ensembl (release 23) genes were used to retrieve orthologs in *Pan troglodytes* (if not available, then from *Pan paniscus*). Ensembl orthologous sequences were aligned using CLUSTAL_W on coding portions of the cDNAs. Small edits were introduced to facilitate the computations. Tables in *Figure 1—source data 1* and *Figure 3—source datas 1* and *2* list the annotated coding genes for the CPR of chromosomes X, 11 and 8, respectively. Estimates of pairwise average nonsynonymous and synonymous divergence and diversity (expected heterozygosity) are based on method I of *Nei and Gojobori (1986)* and *Aguadé et al. (1992)*. 95% confidence limit for estimates of diversity were based on bias corrected bootstrapping (R package *bootstrap* v2017.2). Estimates of the average divergence were derived from the UPGMA clustering. In cases where significant numbers (>10) of apparent recombinants between cenhaps were observed, these were identified and filtered from downstream cenhap group analyses of CDS divergence, expected heterozygosity and TMRCAs (*Figure 1—figure supplement 2* and *Figure 3—figure supplement 2*).

## Estimation of TMRCA for cenhaps

To assign estimates of the ages of TMRCAs the MRCA of each gene in *Homo* and *Pan* was assumed to be 6.5 MYA (*Dutheil et al., 2015*). The estimates of TMRCAs of various cenhaps were calculated from the height of the relevant node on the UPGMA dendrograms based on the average divergence. Approximate confidence intervals of the TMRCAs were estimated as ±two standard deviations in the observed variation across genes in the estimated TMRCA at each node.

## Chromosome-specific array size estimates

Array size estimates were generated using the publicly available mapping of 1000 Genomes sequencing data to GRCh38, including models of CEN regions for each chromosome (*Zheng-Bradley et al., 2017*). For each sample, counts were computed for reads mapping to CEN regions, either uniquely or to multiple sites on a specific chromosome. Chromosome-specific read counts were then normalized by the mean coverage of chr1 unique regions for the sample. We observed significant sample-to-sample variation in array size across chromosomes. Such variation might arise from differences in library preparation, sequencing technology and sequencing center for 1000 Genomes samples. To moderate this issue, we performed a second normalization, dividing estimated array sizes for each chromosome within a sample by the sample sum of array size over all chromosomes.

## HuRef cenhaps

The CEN models incorporated into GRCh38 are based on the long Sanger reads from the HuRef genome (*Levy et al., 2007*). To evaluate the potential impact of reference bias on the estimation of

CEN-mapping of the 1000 Genome Illumina reads for various cenhaps, we determined the similarity of the genotype of the HuRef genome for SNPs defining the cenhaps on chrX and chr11. Since HuRef is a male genome, this involved simply reading off the genotypes at the defining SNPs for chrX. For the autosomes, the diploid genotype of the HuRef genome is needed. Fortunately *Mu et al. (2015)* extended, improved and validated genome-wide genotyping of HuRef. Since the HuRef diploid SNP genotypes are not phased, we attempted to identify the two most likely chr11 cenhap genotypes in HuRef by counting the numbers of mismatches between the diploid HuRef genotypes and each of 2504 individuals in the 1000 Genomes. One individual showed the fewest number of mismatches in the 8816 genotyped sites (chr11:50509493–55326684): seven SNPs exhibited a two-allele mismatch, 165 SNPs a one-allele mismatch (0/0 versus 0/1 or 1/1 versus 0/1) and 8644 SNPs matched. Placed in the context of the seven most common chr11 cenhaps, this individual is a cenhap-heterozygous genotype 3_5 (see *Figure 4—figure supplement 1*). As a group, individuals heterozygous for these two cenhaps, 3 and 5, exhibit the lowest numbers of mismatches from HuRef. *Figure 4—figure supplement 1* shows the distribution of the sums of non-matching SNPs between HuRef and individuals with different cenhap genotypes.

## Acknowledgements

We thank Benjamin Vernot for help accessing archaic DNA sequence data, and Graham Coop, Mikkel Schierup and Yuh Chwen Grace Lee for helpful discussions.

## Additional information

### Funding

| Funder | Grant reference number | Author |
|---|---|---|
| National Institute of General Medical Sciences | R01 GM117420 | Gary H Karpen |
| National Institute of General Medical Sciences | R01 GM119011 | Gary H Karpen |

The funders had no role in study design, data collection and interpretation, or the decision to submit the work for publication.

### Author contributions

Sasha A Langley, Conceptualization, Software, Formal analysis, Investigation, Visualization, Methodology, Writing—original draft, Writing—review and editing; Karen H Miga, Conceptualization, Investigation, Methodology, Writing—review and editing; Gary H Karpen, Conceptualization, Supervision, Investigation, Writing—review and editing; Charles H Langley, Conceptualization, Resources, Formal analysis, Investigation, Visualization, Methodology, Writing—original draft, Writing—review and editing

### Author ORCIDs

Sasha A Langley (D) https://orcid.org/0000-0002-5138-7394
Karen H Miga (D) https://orcid.org/0000-0002-3670-4507
Gary H Karpen (D) https://orcid.org/0000-0003-1534-0385
Charles H Langley (D) https://orcid.org/0000-0001-6160-5503

### Decision letter and Author response

Decision letter https://doi.org/10.7554/eLife.42989.028
Author response https://doi.org/10.7554/eLife.42989.029

## Additional files

### Supplementary files

- Transparent reporting form

DOI: https://doi.org/10.7554/eLife.42989.026

## Data availability

All data needed to evaluate the conclusions in the paper are present in the paper or the supplementary materials. The human populations genomic variation analyzed for linkage disequilibria and haplotypic structure in the Centromere-Proximal Regions of each chromosome was accessed from the 1000 Genomes Project (Phase 3) (ftp.1000genomes.ebi.ac.uk/vol1/ftp/phase3/data/). The inference of Neanderthal and Denisovan ancestry in the Centromere Proximal Regions was based on data available at https://bioinf.eva.mpg.de/jbrowse described in Prüfer et al. 2017. The inference of haplotypes of variation in the Centromere Proximal Regions of the HuRef reference genome used to create the CEN regions in hg38 was based on the genotyping available at http://bioinform.github.io/huref-gs/ and described in Mu, et al. 2015.

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
