## [Decision Letter]

Thank you for submitting your article "Haplotypes spanning centromeric regions reveal persistence of large blocks of archaic DNA" for consideration by *eLife*. Your article has been reviewed by two peer reviewers, including Magnus Nordborg as the Reviewing Editor and Reviewer #1, and the evaluation has been overseen by Diethard Tautz as the Senior Editor. The following individuals involved in review of your submission have agreed to reveal their identity: Andrew G Clark (Reviewer #2).

The reviewers have discussed the reviews with one another and the Reviewing Editor has drafted this decision to help you prepare a revised submission.

Summary:

This is a fascinating and focused paper confirming that the extremely low recombination rate in centromeric regions have led to the preservation of extremely long haplotypes, some of which appear to be of archaic origin. The results are novel, potentially important, and not obvious (for example, the existence of ancient polymorphisms argues against massive segregation distortion and meiotic drive, as you note). The results also suggest interesting avenues for future research. For example, the observation that these haplotypes capture large numbers of odorant receptors suggest a possible role for selection, and the potential importance of centromere driven meiotic drive can also now be explored. In general, it is a significant advance in our understanding of these hitherto inaccessible regions.

Essential revisions:

Although we believe you do an excellent job supporting the conclusions of the paper, we are also well aware of the technical pitfalls inherent in analyzing these kinds of polymorphism data. We're in particular worried about unpredictable interactions between the 4g_dco test and imputation, but there could be other problems that are very difficult to predict and detect. Based on your description of your methods, you are clearly aware of this, and we agree with your logic. Nonetheless, it would be nice to have an independent sanity check, and we would therefore suggest that you confirm the transmission of at a least a subset of these haplotypes using publicly available trio data (1000 Genomes has a few). As you know, this is commonly done in SNP calling, and is an important safeguard against artifactual SNP calls due to misalignment. It will be exciting one day, when there is enough trio data out there, to assess segregation ratios of these cenhaps.

A second point concerns the writing and claims. Arguably, your results are entirely consistent with an entirely neutral process. Is this correct, or is there anything in your data that is incompatible with a standard neutral model? Low recombination gives you long haplotypes that look diverged, but are actually just more visible because of the high LD (as anyone who has looked at *Arabidopsis* data knows). Furthermore, we already know that SNPs are shared with Neanderthals – why would long haplotypes behave differently?

We feel it is important to explicitly discuss whether these haplotypes have a different history than neutral SNPs, and, if there is no evidence for this, change the writing accordingly. Not doing so will likely to lead to misunderstanding and exaggerated claims ("Neanderthal centromeres invaded modern humans!").

[Editors' note: further revisions were requested prior to acceptance, as described below.]

Thank you for resubmitting your work entitled "Haplotypes spanning centromeric regions reveal persistence of large blocks of archaic DNA" for further consideration at *eLife*. Your revised article has been favorably evaluated by Diethard Tautz (Senior Editor) and a Reviewing Editor.

The manuscript has been improved but there are some remaining issues that need to be addressed before acceptance. We were curious why one would not include the trio analysis in the actual paper as well as in the response letter. Perhaps this could be added to Materials and methods section?

---

## [Author Response]

Essential revisions:Although we believe you do an excellent job supporting the conclusions of the paper, we are also well aware of the technical pitfalls inherent in analyzing these kinds of polymorphism data. We're in particular worried about unpredictable interactions between the 4g_dco test and imputation, but there could be other problems that are very difficult to predict and detect. Based on your description of your methods, you are clearly aware of this, and we agree with your logic. Nonetheless, it would be nice to have an independent sanity check, and we would therefore suggest that you confirm the transmission of at a least a subset of these haplotypes using publicly available trio data (1000 Genomes has a few). As you know, this is commonly done in SNP calling, and is an important safeguard against artifactual SNP calls due to misalignment. It will be exciting one day, when there is enough trio data out there, to assess segregation ratios of these cenhaps.

This is a reasonable request. We were, of course, concerned that ShapeIt2 and the way it was parameterized might have been tuned for distinctly different nature of human population genomic variation in the non-Centromere-Proximal regions – the vast majority of and (to most) the more interesting part of the genome. While we became convinced that our success in identifying cenhaps could only mean that ShapeIt2 phasing was robust to such changes in genomic scale of linkage disequilibrium, we surmised that the inclusion of these very trios in the imputation must have contributed critically to the quality of the inference. In response to your request we have gladly conducted a ‘sanity check’. As is documented below the *imputed* centromeric haplotypes in the 1000 Genomes trio parents are, indeed, transmitted intact to the progeny.

While the parents in the 1000 Genomes trios were extensively sequenced, and their entire genome imputed, the available genotyping of the progeny in the majority of these trios is based solely on genotyping arrays (no sequencing or imputation). We focused on data from the OMNI genotyping platform, which includes 353 progeny of 1000 Genomes Phase 3 parents and incorporates light coverage of the CPRs. Thus, data for each trio include a modest number of unphased diploid SNPs in the progeny and the four imputed phased genomes in the two parents. A simple test of the ‘general’ validity of phasing (not a rigorous estimation of a very small error rate) is to consider the predicted four possible genotypes of the progeny. Since our article highlighted variation on chromosomes 8, 10, 11 and 12, we chose to examine the transmission of imputed centromeric haplotypes in these CPRs. If the phasing is correct, then the distributions of the numbers of SNP genotype non-matches between each of the four possible progeny genotypes and the observed progeny’s genotype should be disjunction with one ‘matching’ and the other three exhibiting a number of mismatches. In Author response image 1, one sees a comparison of the distribution over all 353 chr11 trios for both the mean numbers of mismatches (over the four possible genotypes) and that of the minimum of those the four. Trios in which the means were small, ≤ 3 (little power) were excluded. For the vast majority of the remaining 296 trios, the diploid genotype for SNPs in the chr11 cenhap region were indeed transmitted as expected. “Sanity checks out”. Similar analyses and plots for SNPs in the CPRs of chr8, chr10 and chr12 yield the same conclusion.

A second point concerns the writing and claims. Arguably, your results are entirely consistent with an entirely neutral process. Is this correct, or is there anything in your data that is incompatible with a standard neutral model? Low recombination gives you long haplotypes that look diverged, but are actually just more visible because of the high LD (as anyone who has looked at Arabidopsis data knows). Furthermore, we already know that SNPs are shared with Neanderthals – why would long haplotypes behave differently?We feel it is important to explicitly discuss whether these haplotypes have a different history than neutral SNPs, and, if there is no evidence for this, change the writing accordingly. Not doing so will likely to lead to misunderstanding and exaggerated claims ("Neanderthal centromeres invaded modern humans!").

We have further qualified our statements and make no claim about selection’s impact on the distribution of archaic cenhaps. We continue to favor the use of “persistence” in the title because it is correct. But for some readers it may have a broader connotation.

The rigorous analysis of the fit to a selectively neutral demographic model of the genomic variation in many centromere proximal regions, CPRs, in the 1000 Genome data is beyond the scope of this article. The discovery of large-scale haplotypes spanning the vast α-satellite DNA arrays and the introgression of putatively archaic CPRs is in itself a significant observation. The population genetics model-based analysis of such a large and complex data set in the context of the already large and evolving literature of human demographic history is an important challenge for the future. What we present here already is a great deal of work. We feel science will advance more rapidly if we first get the existence of cenhaps in front of the community, since they can be the foundation for many other investigations in more appropriate data sets.

From a different perspective, we note the speculative, yet widely cited/repeated literature expounding the hypothesis that directional selection via meiotic drive and selection for modifiers thereof as an explanation for rapid concerted evolution of centromeric satellite DNAs and rapid divergence of proteins that are primary components of centromere formation/structure/function. Clearly the naïve prediction that recent selective sweeps will wipe out all sequence population genomic diversity in CPR is not supported by our results. But this speculation is not a quantitative model. The time scale on which such sweeps occur could be much greater than the TMRCA of hominins. For those who imagined frequent recurrent meiotic drive sweeps our results are a caution – they are as polymorphic as the non-CPR region. For those who see a likely more complex evolutionary dynamic here, our results hopefully suggest interesting ways to test for their impacts on different time scales.

[Editors' note: further revisions were requested prior to acceptance, as described below.]

The manuscript has been improved but there are some remaining issues that need to be addressed before acceptance. We were curious why one would not include the trio analysis in the actual paper as well as in the response letter. Perhaps this could be added to Materials and methods section?

Thank you for handling our manuscript and helping to improve it. The genomic phasing achieved in the 1000 Genomes (phase 3) explicitly utilizes the trio data as input to the ShapeIt imputation pipeline. For this reason, it would be unexpected that the requested additional analysis would discover inconsistencies. Nevertheless, we did examine the four focal chromosomes for inconsistencies between the reported imputed parental cenhaps and the observed diploid genotypes of the trio progeny as previously reported in our "Author response". Although hundreds of trios had sufficient diversity in the cenhap regions to readily detect phasing errors (and/or genetic exchange), we found only one inconsistent cenhap-trio out of 1294 examined. Thus, we confirm that the phasing of these four chromosomes in the 1000 Genomes (phase 3) is robust in the centromere proximal regions. This conclusion is in agreement with our results: that is, if such trio-based inconsistencies were indeed common they would yield substantial phasing errors and we would not expect to define the clear clustering/descent of cenhaps that we report in our paper. In summary, our focused study of trio data from four chromosomes revealed no obvious discrepancies between the ShapeIt phased parental cenhap genotypes and observed progeny genotypes. We believe that these confirmative results of the 'sanity-check' of the phasing reported in the published 1000 Genomes paper is most appropriately presented in the online correspondence ('Author response') where it can potentially be helpful to concerned readers.